# Effect of Post-Ruminal Urea Supply on Growth Performance of Grazing Nellore Young Bulls at Dry Season

**DOI:** 10.3390/ani13020207

**Published:** 2023-01-06

**Authors:** Irene Alexandre Reis, Mailza Gonçalves de Souza, Yury Tatiana Granja-Salcedo, Isabela Pena Carvalho de Carvalho, Marco Aurélio De Felicio Porcionato, Laura Franco Prados, Gustavo Rezende Siqueira, Flávio Dutra De Resende

**Affiliations:** 1Department of Animal Sciences, São Paulo State University “Júlio de Mesquita Filho” (UNESP), Jaboticabal 14884-900, SP, Brazil; 2Corporación Colombiana de Investigación Agropecuaria (AGROSAVIA), Centro de Investigación El Nus, San Roque, Antioquia 053030, Colombia; 3Trouw Nutrition R&D, 3811 Amersfoort, The Netherlands; 4Trouw Nutrition R&D, Campinas 13080-650, SP, Brazil; 5Agência Paulista de Tecnologia dos Agronegócios (APTA), Colina 14770-000, SP, Brazil

**Keywords:** performance, efficient nitrogen utilization, non-protein nitrogen, post-rumen supplementation

## Abstract

**Simple Summary:**

A fundamental step to meet the growing demand for animal protein and address environmental management is to identify and enhance the production of grazing cattle, that is, to improve the efficiency of use by animals. Efforts to improve nitrogen use efficiency and fiber digestion have focused on improving fiber quality, mainly in low-quality forages, with a post-ruminal delivery source. This approach has aimed to maximize microbial synthesis and reduce losses by excretion. The objective of this study was to evaluate the effect of post-ruminal urea, compared to conventional urea, on the metabolism and performance of Nellore cattle reared on pasture during the dry period. The use of a post-ruminal source presents a delay in relation to the rumen in the delivery of nitrogen through recycling. Therefore, the delivery of ammonia occurs more slowly and steadily throughout the day, which would result in a delay. Our findings highlight differences only in crude protein intake, supplement, and protein digestibility for post-ruminal urea production.

**Abstract:**

The objective of this study was to evaluate the effect of the use of post-ruminal urea on performance, nitrogen metabolism and the ruminal environment of Nellore cattle reared on pasture during the dry season. In experiment 1 (Exp. 1), nine ruminal-cannulated Nellore steers, 30 ± 2 months old (651 ± 45 kg body weight (BW)), were allotted to a 3 × 3 Latin triple square. In experiment 2 (Exp. 2), 84 Nellore bulls, 18 ± 3 months old (315 ± 84 kg BW), were distributed in complete randomized blocks, by initial BW. Protein supplements were supplied daily, in the amount of 2 g/kg BW, and consisted of either CONT: protein + conventional urea (50% CP), PRU: protein + post ruminal urea (50% CP) and U + PRU: protein + urea conventional + post-ruminal urea (70% CP). The paddocks were composed of *Urochloa brizantha* cv. Marandu grass. In Exp. 1, there was no treatment effect for DM, OM, NDF, forage intake, and CP, but there was a higher intake for PRU (*p* < 0.005) and a higher digestibility for U+ PRU (*p* = 0.001). There was no effect on ruminal pH or NH_3_-N concentration (*p* ≥ 0.232), but there was an interaction between treatment and time for them (*p* < 0.039). Furthermore, there was a treatment effect on the total SCFA concentration, with CONT being higher than the others. A difference in the acetate:propionate ratio was found (*p* < 0.027), with a greater relationship for PRU and U + PRU. A treatment effect (*p* = 0.049) was found for the propionate proportion, with a higher proportion in the CONT. Nitrogen intake was consequently lower for the CONT and higher urinary excretion for the U + PRU (*p* = 0.002). Animals supplemented with CONT showed a tendency to have more Bacteria and fewer Archaea (*p* = 0.086). In Exp. 2, there was a treatment effect for the disappearance rate of the supplement (*p* < 0.001). Intake was faster for PRU and CONT, but performance was not affected by PRU (*p* = 0.311). The use of post-ruminal urea alters the microbial population, but does not affect performance. Therefore, supplementation with post-ruminal urea presented similar results compared to conventional urea. Ruminal and blood parameters and animal performance were not influenced by treatments.

## 1. Introduction

There are growing environmental concerns regarding cattle production systems, since they are considered one of the main contributors to the losses of nitrogen (N) in the environment. Ruminants have a low efficiency of nitrogen assimilation from the diet (around 25%), due to the large loss of ammonia in the rumen. This means that most of the protein supplied is excreted in urine and feces [1,2]. Low efficiency has implications in both animal performance and the environment. However, these animals have developed a remarkable ability to reuse excess urea as an available source of nitrogen [3], particularly during periods of dietary protein deficiency [4]. They are also capable of extracting maximum amounts of nutrients from low-quality fibrous feeds.

During the dry season, forage content increases in fiber, mainly due to development, which results in a reduction in both crude protein and soluble carbohydrate contents [5,6]. This results in nutritional limitations and insufficiencies of available diet basal resources. Since fibrous carbohydrate fermenting bacteria use ammonia as a nitrogen source, the use of non-protein nitrogen (NPN) sources, such as urea, represents an option to meet the animal’s protein requirements [7].

However, urea, which is widely used as a source of nitrogen in ruminant nutrition, is rapidly hydrolyzed in the rumen, allowing limitations on feed intake and harmful effects on feed digestibility where the high load of ammonia released in the rumen increases the transport of ammonia into the blood [3]. Consequently, more ammonia is absorbed before being used by ruminal microorganisms, and if absorption exceeds the animal’s ability to recycle urea back into the rumen, nitrogen is lost through urinary excretion [3,8]. Moreover, there is a negative impact on animal performance due to the negative effects on intake [8].

Nitrogen can be considered the main component of supplements for grazing cattle, especially during the dry season, and its utilization efficiency is an important parameter to define the composition of supplements, and to understand the efficiency of animal production [5]. A major challenge is to improve the understanding of nitrogen metabolism to formulate the most efficient diets and improve the nutritional management of grazing cattle [8].

Environmental concerns include the volatilization of ammonia in the animal’s excreta and methane produced from ruminal fermentation. Therefore, it has been established that excess nitrogen in the environment can have adverse effects. In this context, the rate of dietary protein degradation in the rumen can be reduced to improve carbohydrate utilization and thereby improve efficiency and nitrogen retention [2]. One of the main implications of the production system is the efficient use of nutrients. Inefficient use results in losses, as well as economic damage. Thus, adapting the relationship between the quantity and quality of protein required by the animal, combined with increased productivity, brings benefits to the efficient use of N [9]. In this case, reducing N losses is essential for greater use of feed and microbial synthesis of ruminant animals.

In addition, the site of digestion and absorption of N sources could influence the N excretion, urea recycling, and the efficiency of use of N by the animal [10,11]. In addition, understanding the possibilities of N delivery with a post-ruminal source, to balance the supply of nutrients to enhance digestion and utilization, is a major challenge, but also a supplementary strategy to improve livestock production, further reducing plus the resource used for animal production and the elimination of N into the environment [11,12]. The use of a post-ruminal urea source presents a delay in relation to the rumen in the delivery of N through recycling, thus the delivery of ammonia occurs more slowly and steadily throughout the day. Considering a forage of low quality, the slow supply of ammonia in the rumen environment would allow the microorganisms to extract energy from the basal substrate in the best way, which would result in greater efficiency by increasing microbial synthesis, which may imply a greater supply of metabolizable protein (MP) [11].

Thus, this study aimed to evaluate the effect of post-ruminal urea use on performance, N metabolism and the ruminal environment of Nellore cattle raised on pasture during the dry period. Our hypothesis is that the use of NPN becomes more efficient when using an available post-ruminal source, as this does not result in animal intake restrictions, as well the combination of sources available in the rumen and post-rumen, due to the increase in the supply of metabolizable protein, results in greater nitrogen utilization efficiency (recycling), better fiber digestibility, more stable rumen fermentation parameters, through microbiota manipulation, thus promoting better results in the performance of cattle on pasture in the dry period.

## 2. Materials and Methods

### 2.1. Location

This study was conducted at the Agência Paulista de Tecnologia dos Agronegócios (APTA), Alta Mogiana regional pole, Colina, São Paulo, Brazil, from June 2020 to October 2020 during the dry season. The climate is subtropical humid, characterized by dry winters and rainy summers. During the experiment, the total precipitation was 27.8 mm (totalizing 6 rainy days). The study was conducted in accordance with animal welfare guidelines and the protocol was approved by the Ethics, Bioethics, and Animal Welfare Committeeof the *Universidade Estadual Paulista* (UNESP), Jaboticabal campus, Brazil (Protocol number 3974/20).

The study was divided into two experiments: metabolism and performance of Nellore cattle supplemented with post-ruminal urea and conventional urea. The experiments were conducted simultaneously. In experiment 1 (Exp. 1), the evaluation of animal intake, apparent total tract digestibility, ruminal and blood parameters, nitrogen balance, and ruminal bacteria was conducted. In experiment 2 (Exp. 2), the evaluation of animal intake, apparent total tract digestibility, and ruminal parameters was conducted.

### 2.2. Animals, Area and Experimental Design

Experiment 1 lasted 63 days, divided into three periods of 21 days each for evaluation and data collection. Nine ruminally cannulated Nellore steers [651 ± 45 kg of body weight (BW) and 30 ± 2 months old] were used in a triple 3 × 3 Latin square design, with three periods and three treatments. Each period represented an experimental unit. The animals in this experiment were very heavy animals. The steers were distributed into three paddocks of *Urochloa brizantha* cv. Marandu at approximately 1.0 ha each. Each paddock contained water troughs and feed bunks. The animals were adapted, and after this period, the fecal excretion markers stabilized in six days. On days 15–18, fecal samples were collected to estimate forage and supplement intake. Urine samples were collected on days 17–18 in order to estimate the nitrogen balance. On day 19, ruminal samples were collected for digestive and fermentative parameter analysis, ruminal microbial diversity was collected on day 21, and blood samples were collected on days 20 and 21.

Experiment 2 lasted 120 days, from 16 June to October 2020. During the first 8 days, animals were adapted to the environment, and performance was evaluated in four periods of 28 days each. Eighty-four Nellore bulls (315 ± 84 kg of BW and 18 ± 3 months old) were categorized by BW and distributed in 12 paddocks (six to eight animals per paddock) in a randomized blocks design with 3 treatments and 4 blocks, where each paddock represented an experimental unit. The paddocks were composed of *Urochloa brizantha* cv. Marandu with 6 paddocks of 3.4 ha and 6 paddocks of 4 ha containing water troughs and feed bunks.

### 2.3. Treatments

The treatments consist of three supplements with the difference being the NPN source (urea or post-ruminal urea or both sources with a higher level of crude protein. The control (CONT) protein supplement with 11.3% conventional urea with 50% crude protein (CP) (Lambisk S—commercial product of Bellmann-Trouw Nutrition). The post-ruminal urea (PRU) protein supplement contained 12.7% post-ruminal urea (NPN equivalent/conventional urea) with 50% CP, and the combination (U + PRU)-protein supplement contained 11.3% of conventional urea and 8% of post-ruminal urea, increasing the level of NNP with the inclusion of post-ruminal getting with 70% CP. The animals of the two experiments were fed daily at 9 am, with protein supplements offered at 0.2% BW (Table 1). Before supplying the supplement each day, the leftovers from the previous day’s supply were evaluated. Leftovers were considered to be 5% of the supplied quantity and adjustments were made at the end of each period based on the last weighing of the animals.

The post-ruminal urea was coated with an in vitro ruminal release of 6% and in vitro digestibility of 94%. The method used to evaluate the in vitro product can be found in Appendix B.

### 2.4. Forage Evaluation

Every 28 days, the mass was analyzed for the two experiments by using the double sampling method [13]. The quantitative and structural components of the forage were evaluated with samples at the average height of each paddock, divided into fractions. To estimate the nutritional value of forage, hand-plucked samples were collected [14]. For chemical analysis, these samples were partially dried in a forced-air circulation oven set at 55 °C for 72 h, ground through a Wiley mill in 2- and 1-mm sieves. The continuous stocking rate (put and take) was used as the grazing method [15]. The quantitative components and chemical composition of the forage (Experiment 1 and Experiment 2) are presented with average values in Table 2. In Experiment 2, the averages of the experimental periods were characterized, because of the variation in quality during the dry season.

For the chemical composition of supplements, forage, and feces the DM content (method 934.01), mineral matter (MM; method 942.05), crude protein (CP; method 978.04), and ether extract (EE; method 920.39) were used, according to AOAC [16]. The neutral detergent fiber (NDF) with described by Robertson and Van Soest [17], using a Tecnal^®^ TE-149 fiber analyzer. To determine the indigestible neutral detergent fiber (iNDF) (iNDF) the hand-plucked samples with 2 mm were incubated in the original location for 288 h as described by Valente et al. [18].

### 2.5. Intake and Digestibility

Forage and supplement intake were estimated in each period using markers. To estimate fecal excretion, supplement, and forage intake, chromium oxide (Cr_2_O_3_) titanium dioxide (TiO_2_), and indigestible NDF (iNDF) were used. For 10 days, 10 g per animal/day of (Cr_2_O_3_) was placed directly in the rumen, (6 days before sampling and 4 days during the fecal excretion collection period, from day one. Fecal samples were collected once daily and alternated at the following times: 7 am, 10 am, 1 pm, and 4 pm. Fecal samples were weighed and partially oven-dried at 55 °C for 72 h and ground. A sample was collected from each animal in each sampling period and stored for analysis.

Fecal excretion was calculated according to the following equation: fecal excretion = [chromium oxide supplied (g/day)]/[fecal chromium oxide concentration (g/g MS)] [19]. To estimate the dry matter intake (DMI) of the supplement, TiO_2_ was used at 10 g/day per animal, for 10 days as described for Titgemeyer et al. [20]. Fecal samples were collected simultaneously with the fecal excretion procedures. Fecal samples were digested using sulfuric acid and analyzed as described by Myers et al. [21]. Individual supplement intake was estimated using the following equation: DMI supplement = [g of TiO_2_/g of feces × fecal excretion g/d]/[gTiO_2_/g of the supplement]. The forage DMI was estimated using the iNDF market, determined after ruminal incubation [18], using 2 mm ground forage samples. Forage DMI was estimated from the fecal output of the internal marker corrected for the supplement contribution as follows: Forage DMI = [fecal excretion g/d × (iMF) − DMI of supplement × (iMS)]/[iMH]. iMF, iMS, and iMH are the concentrations of the internal marker in the feces, supplement, and forage.

In experiment 2, the rate of the supplement in the trough was monitored halfway through each experimental period, every 1, 3, 5, 8, and 24 h after being offered, in order to evaluate the intake behavior of the animals concerning the treatments.

### 2.6. Nitrogen Balance

Urine collection was performed on the 18th day of the experimental period in the cannulated animals, approximately 0, 3, 6, and 12 h after supplementation. Samples were collected in the spot form, by urination stimulated by urethral massage [22]. Samples of 10 mL were diluted with 40 mL of sulfuric acid (0.036 N) to quantify allantoin concentrations by the colorimetric method described by Chen and Gomes [23]. The reading was performed on an ASYS^®^ microplate reader; creatinine (cod K-222) and uric acid (cod K-139), were performed using commercial kits (Bioclin^®^). The readings were performed in an automatic biochemistry analyzer (Sistema de Bioquímica Automático SBA-200; CELM^®^). Total urine volume was estimated by dividing the daily urine output by the creatinine concentration, as described by Costa e Silva et al. [24]. The total nitrogen urine concentration (method 978.04).

The absorbed purines (X, mMol/day) were calculated from the excretion of purine derivatives (Y, mMol/day) using the equation described by Chen and Gomes [23], as follows: Y = 0.85X + (0.385 × kg BW^0.75^), in which 0.85 is the recovery of purines absorbed as purine derivatives in the urine, and 0.385 endogenous contribution to purine excretion. The synthesis of microbial nitrogen (Nmic), (Y, gN/day) was calculated as a function of the absorbed purines (X, mMol/day), using the formula Y = 70X/0.83 × 0.116 × 1000, in which 70 is the purine nitrogen in mg N/mMol; 0.83 was the digestibility of microbial purines and 0.116 was the ratio of purine N: Total N of microorganisms, described by Chen and Gomes [23]. Nitrogen balance (NB), expressed in g/day, was obtained by the difference between consumed nitrogen (CN) and nitrogen excreted in feces (NEF) and nitrogen excreted in urine (NEU) in g/day. The concentration of total N in the feces was also determined (method 978.04) according to the AOAC [16]. The N utilization efficiency was calculated by dividing the retained nitrogen by the intake.

### 2.7. Ruminal Fermentation Parameters

Composite ruminal samples (dorsal, central, and ventral regions) were collected from each cannulated animal, at 0, 3, 6, and 12 h after supplementation (day 19 and 21) and immediately filtered through two layers of gauze.

NH_3_^−^N was determined at the same times as above, samples of (15 mL) were preserved with 1 mL of H_2_SO_4_ (sulfuric acid) and stored at −20 °C until analysis by the phenol-hypochlorite colorimetric method [25]. The samples were used for rumen pH measurement using an electric pH meter (DM-22, Digimed, São Paulo, Brazil). Samples (15 mL) were stored at −20 °C for short-chain fatty acid (SCFA) concentration analysis (acetate, propionate, butyrate, and valerate). Samples were centrifuged at 15,000× *g* × 15 min at 4 °C (Sorvall Superspeed RC2-B, Newton, CT, USA). All processing of samples and analysis was carried out according to the method described by [26]. The calibration curve was performed with chromatographic standards (Chem Service) as reported in the work by Cidrini et al. [27].

A total of approximately 50 g per animal (comprising a mix of liquid and solid) was collected through the ruminal cannula on day 21 of each period, 3 h after supplementation, and immediately stored at −80 °C until further analysis. The samples were processed to obtain a bacterial pellet [28]. A Quick-DNA™ Fecal/Soil Microbe Miniprep kit extraction was used to extract metagenomic DNA from 250 mg of bacterial pellet according to the manufacturer’s instructions (Zymo Research Corporation, CA, USA); a FastPrep-24 Classic Instrument (MP Biomedicals, France) was used to lyse cells, and both DNA yield and DNA quality were evaluated as described by Granja-Salcedo et al. [28].

Duplicate libraries were prepared by PCR amplification of the V3 and V4 regions of the 16S ribosomal RNA gene (16S rRNA) for bacteria using the universal primers 515F (5-GTGCCAGCMGC CGCGGTAA-3) and 806R (5-GGACTACHVGGGTWTCTAAT-3) as described by Caporaso et al. [29]. PCR fragments were purified using the Zymoclean Gel DNA Recovery kit, following the manufacturer’s instructions. The resulting fragments were submitted to sequencing on an Illumina NovaSeq6000 PE 250 platform, resulting in an average of 160.000 reads per sample. Reads were mapped against a reference 16S rRNA database (Silva 138 99% OTUs from 515F/806R region of sequences). Sequence trimming was performed by selecting sequences over ~470 bp in length with an average quality score greater than 40 based on Phred quality, and duplicate reads were removed using the Prinseq program [30]. Quantitative Insights into Microbial Ecology (QIIME) software package version (2022.2.0) was used to filter reads and determine operational taxonomic units (OTUs) as described by Cole et al. [31]. Significant readings were classified based on the Multinomial Naive Bayes algorithm to cluster the reads OTUs with a 99% cutoff and to assign taxonomy, Silva 138 99% OTUs from 515F/806R region of sequences is used. For the analysis of functional categories, a sequence identity cutoff of 97% was applied, and functions were assigned using the Kyoto Encyclopedia of Genes and Genomes (KEGG) database using the picrust2 (v2.4.2).

### 2.8. Blood Parameters

Blood collections were performed by jugular venipuncture with vacuum tubes without anticoagulant (BD Vacutainer^®^) at 0, 3, and 6 h after supplementation over two days (Day 20: 0 and 6 h, and Day 21: 3 h). The samples were centrifuged at 3080× g for 15 min at 4 °C. Serum was harvested and stored at −20 °C until later analysis. The serum was analyzed for uric acid, urea, creatinine, albumin, total protein, and liver enzymes (aspartate aminotransferase—AST and gamma glutamyltransferase—GGT), for the analysis commercial kits from the company (Bioclin^®^) according to the manufacturer’s specifications, (uric acid, code K-139), (urea, code K-056), (albumin, code K-040), (creatinine, code K-222), (total protein, code K-031), (AST, code K-048) and (GGT, code K-060). The readings were performed in an automatic biochemistry analyzer (Sistema de Bioquímica Automático SBA-200; CELM^®^).

### 2.9. Animal Performance

To calculate the average daily gain (ADG) of the animals, weighing was performed at the beginning initial body weight (IBW) and (final body weight (FBW) of the experimental period, after a 16 h feed and water fasting. The procedure described above was performed at the beginning and at the end of the adaptation period. ADG was calculated using the initial BW and final BW of the experimental period divided by the number of days evaluation period.

### 2.10. Statistical Analyses

Data on forage characteristics, intake, digestibility, and nitrogen metabolism in experiment 1 were analyzed between treatments by ANOVA using a 3 × 3 triple Latin square design (three treatments and three periods), considering the animal versus period as the experimental unit, by using the PROC MIXED of SAS (SAS Inst. Inc., Cary, NC, United States). The variables conducted on the same animal, but at different times, were analyzed as a repeated measure over time (pH, NH_3_^−^N, SCFA, blood parameters). All outlier animals were removed for data analysis.

Estimates of richness, diversity index and relative rumen microbial abundance (Bacteria and Archaea) were compared between treatments using the Kruskal–Wallis test. To compare significant medians, Dunn’s post hoc test was used. The principal component analysis (PCA) was used to extract important microbial OTUs associated with parameters of rumen fermentation, ingestion, digestibility, and the KEGG pathway considering the supplement (treatment) using the Factor Miner package in R.

The data were analyzed using a randomized block design with three treatments and four replications, with each paddock considered an experimental unit. A mixed model was used and included treatments as a fixed effect and the block as a random effect. The data obtained over time were analyzed as repeated measures, adding the effects of period and the interaction between period and treatment in the model. The data were analyzed using the PROC MIXED of SAS (SAS Inst. Inc., Cary, NC, United States), with a previous normal distribution test (Shapiro–Wilk test) and homoscedasticity of variances (Bartlett test). The lowest value of BIC (Bayesian information criterion) was used to determine the matrices chosen for each variable, in the parameters analyzed over time. Significance was considered when *p* ≤ 0.05, while trend was considered when 0.05 < *p* ≤ 0.10 for all tests.

## 3. Results

### 3.1. Intake and Apparent Digestibility

No effects were registered for intake of DM, OM, forage, and NDF (*p* > 0.454). A treatment effect for protein intake (*p* = 0.005) was noticed, higher for PRU and U + PRU treatments, and supplement intake (*p* < 0.001). The effect was greater on the PRU treatment than the others (Table 3). No effects for apparent digestibility of DM, OM, and NDF (*p* > 0.332), except for crude protein (*p* < 0.001), which presented a higher digestibility level for the U + PRU treatments (356 g/kg), followed by PRU (308 g/kg). As for the intake in the body weight percentage, there was a difference only for the supplement intake (*p* < 0.001) higher for PRU (Table 3).

### 3.2. Ruminal Fermentation

There was an interaction between treatment and hour, three hours after supplementation. Animals fed PRU and U + PRU had greater pH (7.04) (*p* = 0.008 and *p* = 0.021) than animals fed CONT (6.88). Six hours after supplementation, all animals had similar pH (6.96) (*p* > 0.701). In addition, twelve hours after supplementation the pH of animals fed PRU tended (*p* = 0.084) (6.73) to be lower compared to U + PRU (6.93) (Figure 1). No treatment effects for ruminal pH were registered (*p* = 0.232), and all treatments had similar pH (Table 4).

NH_3_^−^N data over the experimental periods show no effects (*p* = 0.261) (Table 4). However, the trend of treatment effect and collection time (*p* = 0.002) were found to be significant, before supplementation. The CONT group tended to have a greater treatment effect (*p* = 0.082) than the PRU group, and both groups had similar levels of NH_3_^−^N than the U + PRU group.

There was also treatment–hour interaction for NH_3_^−^N, (*p* = 0,002). Before supplementation, CONT (8.23 mg/dL) treatment tended to be greater (*p* = 0.082) than PRU (5.1 mg/dL) and both similar levels of NH_3_^−^N than U + PRU. Three hours after supplementation, a peak concentration and difference was registered, in which the animals that consumed PRU had a bigger concentration (15.8 mg/dL) (*p* = 0.001) than animals that received CONT (7.45 mg/dL) and the U + PRU (11.8 mg/dL) (*p* = 0.013). U + PRU had a bigger concentration compared to CONT (*p* = 0.013). Six hours later, there was a difference between PRU (14.80 mg/dL) and U + PRU (11.14 mg/dL) (*p* = 0.055). Twelve hours after supplementation, U + PRU treatment showed a higher NH_3_-N (13.85 mg/dL) (*p* = 0.024) content than PRU (9.46 mg/dL), and both were equal to CONT (11.4 mg/dL) (*p* > 0.194) (Figure 1).

Short-chain total fatty acids (SCFA) presented treatment effects (*p* < 0.027). The CONT treatment showed greater amounts of total SCFA compared to the PRU (*p* = 0.036) and U + PRU (*p* = 0.012) treatments. No effects on treatment, time, and treatment versus time interaction effects were observed for almost all variables on individual SCFA production (*p* > 0.182). One exception was the proportion of propionate (*p* = 0.049), in which a higher molar proportion occurred in the CONT treatment than in the others (Table 4). There was a treatment effect for the acetate:propionate ratio (*p* = 0.015), the treatment U + PRU and PRU showed a higher ratio (*p* = 0.043) and (*p* = 0.004) compared to the CONT treatment.

### 3.3. Blood Parameters

The different treatments did not influence the concentrations of blood metabolites (*p* > 0.309). The concentrations of blood parameters are listed in (Table 4). The variables uric acid, urea, albumin, total protein, and AST (*p* < 0.056) presented time-related effects, with an increase in the concentration of metabolites at the time of collection.

### 3.4. Nitrogen Metabolism

Nitrogen (N) intake from forage and total nitrogen were not different (*p* > 0.156), shown in Table 5. However, nitrogen intake from the supplement was different between treatments (*p* = 0.003), higher for the PRU treatment and the combination of sources U + PRU (*p* = 0.001) in relation the CONT. Nitrogen excretion was not affected by treatments (*p* > 0.425), except for the urinary volume (*p* = 0.002). A higher volume was registered for the U + PRU treatment compared to the others, and PRU tended to have a higher volume (*p* = 0.090) than CONT. The treatments also did not affect balance, nitrogen retention, and microbial CP (*p* > 0.219).

### 3.5. Ruminal Microbial Diversity

Illumina sequencing produced 4.401 sequences from the 27 samples. After trimming, the median number of sequences was 37.00 per sample, with a coverage median of 99%. The ruminal microbial population was not affected by treatments (*p* = 0.882). According to PERMANOVA, the population of bacteria was higher for CONT (*p* = 0.086), and the population of Archaea was lower for CONT (*p* = 0.086) (Table 6). The richness index (ACE and CHAO 1) and diversity estimators (Fisher, Simpson, and Shannon Wiener) values were observed based on the post-ruminal urea, conventional urea, or a combination of both and were similar among the treatments (*p* > 0.707) (Table 6). Twenty-two phyla were detected, but only four were influenced by the treatments (Table 6), Bacteroidota and Fibrobacterota were the most abundant phyla in CONT (*p* < 0.062). The abundance of Proteobacteria and Halobacterota was higher in the rumen of U + PRU (*p* < 0.045), and there was an increasing trend for the Firmicutes:Bacteroidetes ratio (*p* = 0.099) for the U + PRU.

A total of 270 families and 486 genera were identified, and 29 families showed differences in relative abundance among the supplementation (Table 7). A higher relative abundance of *Clostridia_vadinBB60_group and Corynebacteriales* (*p* < 0.071) for PRU treatment was registered. *Gitt.GS.136, Desulfobulbaceae, Eubacteriaceae*, and *Nitrospirota* were identified only in the rumen of animals supplemented with PRU. There was a higher relative abundance of the *uncultured* family and *Fibrobacteraceae* (*p* < 0.013) in CONT. Furthermore, the higher relative abundance of *Beijerinckiaceae, Comamonadaceae, Oxalobacteraceae, Devosiaceae, and Methanomicrobiaceae* (*p* < 0.040) in the U + PRU. *Ktedonobacteraceae, Leuconostocaceae, Staphylococcaceae, Pla4_lineage, vadinHA49*, and *Paracaedibacteraceae* were present only in the rumen of U + PRU. Meanwhile, PRU and U + PRU had a higher abundance of *Microbacteriaceae, Rhizobiaceae*, and *Clostridiaceae* (*p* < 0.051). In addition, other families were only identified in the rumen of PRU and U + PRU, such as *Kineosporiaceae, Frankiaceae, Pseudonocardiaceae, Planococcaceae, Rokubacteriales_WX65*, and *Hyphomicrobiaceae* (Table 7).

In addition, families that showed no differences in the treatments were identified only in a specific group, such as lastocatellia__11.24, TRA3.20, uncultured, Entotheonellaceae, Microtrichaceae, Micropepsaceae, (Acidobacteriota) Subgroup_5 and Subgroup_12, Halieaceae, AKYH767, Cytophagaceae, Caldilineaceae, Verrucomicrobiaceae, and Nannocystaceae were identified only in animals supplemented with PRU. Bifidobacteriaceae, Akkermansiaceae, Methylophilaceae, Thermoactinomycetaceae, and Pirellulaceae were present only in CONT. Acidobacteriaceae_Subgroup1, uncultured, Nakamurellaceae, Acidothermaceae, Izemoplasmataceae, Bdellovibrionaceae, and TK10 only in the combination of U + PRU sources. The families Rhodobacteraceae, Fusobacteriaceae, Cryptosporangiaceae, Williamwhitmaniaceae, Streptomycetaceae, Mycobacteriaceae, Morganellaceae, Alcaligenaceae, Xiphinematobacteraceae, and Subgroup_25 were identified in the PRU and U + PRU groups.

At the bacterial genus level, 50 bacterial genera showed variations between the treatments (Appendix A). PRU had higher ruminal relative abundance of *DNF00809, Anaerovorax* and *Clostridia_vadinBB60_group* (*p* < 0.054), and a lower abundance of *uncultured, U29.B03 Endomicrobium,* and *Fibrobacter* (*p* < 0.095). Consequently, a higher abundance of these genera was noticed for CONT. *Kribbella, Gitt.GS.136, Desulfuromonas, Desulfobulbus, Eubacterium, Flavonifractor, Howardella,* and *Caulobacter* were identified only in the PRU. In contrast, the relative abundance of genus *Lachnospiraceae_NK4B4_group, Comamonas, Devosia, Massilia, Pseudomonas*, and *Methanomicrobium* was higher for U + PRU (*p* < 0.079). The genus *Kineococcus, Uncultured, Bdellovibrio, Allobaculum, Peptococcus, Staphylococcus, Weissella, vadinHA49, 1174.901.12, Belnapia*, and *Rubellimicrobium* were present only in the U + PRU rumen. The genus *Actinomycetospora, Jatrophihabi-tans, Kineosporia, Bacteroides_pectinophilus_group, Lysinibacillus, Peptoclostridium*, and *Hyphomicrobium* were identified only in the rumen of PRU and U + PRU. *ADurb.Bin063.1* was identified only in the CONT rumen (*p* < 0.091) Appendix A.

The Euclidean distance of treatments showed a separation among animals supplemented with CONT and those supplemented with PRU or U + PRU (Figure 2). Principal component analysis (PCA) extracted 34 variables that explained 63.2% of the total variability. CONT steers were mainly distributed along the positive region of both principal components 1 and 2 (Dim 1 and 2), with a positive correlation with both families Fibrobacteraceae (r = 0.50) and Prevotellaceae (r = 0.96), and Bacteroidetes phylum (r = 0.94). The total AGCC ruminal (r = 0.37), DNA replication (r = 0.73), the metabolism of pyrimidine (r = 0.78), Ala, Asp, and Glu (r = 0.81), Arg and Pro (r = 0.67), and Amino sugar and nucleotide sugar (r = 0.88), were also positively correlated with CONT.

The PRU steers were mainly distributed along the negative region of dimension 1 and were positively correlated with phyla Firmicutes (r = 0.81), Desulfobacterota (r = 0.74), and Euryarchaeota (r = 0.52), both ratio Firmicutes:Bacteroidetes (r = 0.92) and Acetate: Propionate (r = 0.35), ACE richness index (r = 0.38), Fisher diversity estimator (r = 0.40), the ruminal pH (r = 0.45), the biosynthesis of Val, Leu, and Ile (r = 0.62), and the metabolism of linoleic acid (r = 0.83). While a negative correlation was observed with the biosynthesis and metabolism of glycan (r= −0.82), Nitrogen (r= −0.87), and Carbohydrates (r = −0.85).

The U + PRU steers were distributed mainly in the negative region of dimension 2, and they were positively associated with the phyla Proteobacteria (r = 0.93) and Actinobacteria (r = 0.71), the Peptostreptococcaceae family (r = 0.63), crude protein digestibility (r = 0.43), the degradation of Val, Leu and Ile (r = 0.46) and Lys (r = 0.62), and the metabolism of nitrogen (r = 0.28), carbohydrates (r = 0.85), beta alanine (r = 0.68), Phe (r = 0.71), and associated negatively with the biosynthesis of Lys (r = −0.92) and the metabolism of His (r = −0.76) and Cys and Met (r = −0.84).

### 3.6. Disappearance of the Supplement in the Trough

The supplement disappearance rate was monitored between 1, 3, 5, 8, and 24 h after offered. A treatment effect for the animals that consume supplements with a lower amount of protein (*p* < 0.001), CONT and PRU, presented a higher intake at each time interval, reaching 3% within 24 h after being offered. The treatment with the highest U + PRU protein content showed the lowest intake at each time interval, reaching 37% within 24 h after being offered (Figure 3). Treatment and monitoring time showed an interaction (*p* < 0.001). 1 h after supply, CONT and PRU showed a higher intake level in the time interval (*p* < 0.001) concerning U + PRU. 3 h after, PRU presented a higher intake (*p* = 0.092) than CONT, and both had higher intake (*p* < 0.001) than U + PRU at 5, 8, and 24 h after supplementation.

### 3.7. Animal Performance

No difference was recorded between treatments for BW (*p* = 0.842), with a final body weight of 351 kg. However, there was a difference in the periods (*p* < 0.001) with an increase in the BW over the periods, as per Table 8.

There was an interaction effect between treatment and period for the average daily gain (ADG) (*p* < 0.036). In the first period (0 to 29 days), the PRU treatment tended (*p* = 0.073) to present a greater gain (0.605 kg/d) than CONT (0.522 kg/d), and both were similar to U + PRU. In the second period (29 to 58 days), all animals had a similar ADG (*p* > 0.511). In the third period (58 to 87 days), animals fed CONT had a higher ADG (0.343 kg) (*p* = 0.078) than the other treatments (0.226 kg). In the fourth period, all animals lost weight; however, animals fed PRU lost less weight (−0.025 kg) than animals fed the other supplements (−0.163 kg) (*p* ≥ 0.044) (Figure 4).

## 4. Discussion

During the dry season, forage quality and quantity decrease, which can have detrimental effects on the performance of grazing cattle. Throughout the experimental periods, the structural characteristics of the forage underwent changes. Considering the total forage dry matter availability in this period to be the limiting factor for intake and animal production, it is expected that animal production will decrease as the experimental period continues.

The extent of the animal’s response varies according to the quality of the basal diet. At the end of the first experimental period, there was a total reduction in the amount of green leaves and an increase in the proportion of stem and stem. The high proportion of these components associated with low pasture density may reduce the bite size and consequently allow a lower intake and digestibility. This may influence the effects of supplementation. During periods of dry weather, forage CP levels are usually below 70 g/kg DM [32,33], which was also observed in this study. Critical CP levels are considered to be important for microbial activity in the rumen. Thus, as the high levels of NDF (81.6%), ADF (45%), and lignin (6%) were also observed in this study, the low nutritional value of the forage is related to the CP content, high fiber content and low digestibility. 

The intake of DM, OM, NDF, and forage were not affected by the supplements; this is in line with other studies by Carvalho et al. [12] and Oliveira et al. [11], where the post-ruminal urea did not influence the extent of fiber digestibility, contrary to what was observed by Carvalho et al. [12] with urea infusion in the abomasum. Crude protein intake was higher for the PRU and U + PRU treatments due to the high CP content. Therefore, the supplement intake was higher for the PRU treatment, as the animals did not show intake limitation as in the U + PRU treatments. Urea is one of the main limits of supplement intake due to its low acceptability by animals and high rates of degradation in the first hours after feeding. This leads to excess NH_3_ in the rumen, therefore, high levels of urea reducing supplement intake [34]. Thus, the high-protein content for microbial and animal metabolism can cause hepatic depressions of NADH and NADP related to the elevation of hepatic ammonia level, generating negative effects on carbohydrate metabolism [35].

We observed that when high levels of urea (U + PRU) were delivered, negative consequences were observed for the urinary excretion and intake of the animals. Despite the higher N intake for PRU and U + PRU concerning CONT, the balance of nitrogen and microbial nitrogen was not significant between treatments, as also observed in the study by Carvalho et al. [12]. The higher protein digestibility for the combination of sources (U + PRU), with urea available in the rumen and post-rumen, followed by PRU, was expected, as the apparent digestibility of a non-fibrous compound is positively associated with its intake. This is in agreement with the CP digestibility pattern both in the rumen and in the post-rumen, which was also reported by Oliveira et al. [11], increasing when urea was provided in these compartments.

Blood parameters were similar between the supplements, the metabolic profile being an important indicator of metabolism, which reflected on the balance and mobilization of nutrients in the tissues for both treatments. However, the animals used for the metabolism study were heavier and had a limited supplement intake. Consequently, so these animals did not consume the amount equivalent to their body weight, which may have interfered with the greater accuracy of the metabolic profile. Despite this, most of the blood metabolite profile was within what is considered adequate for animals in this category, according to [36,37], except for creatinine and total protein, which were above what was related by these authors.

Ruminal pH values remained above 6.7 for all treatments, which is considered adequate for microbial growth and the activity of mainly cellulolytic bacteria in the rumen of beef cattle, such as *Ruminococcus* spp. and *Fibrobacter* spp. [38,39]. All treatments showed adequate concentrations of ammoniacal nitrogen according to [40], optimal levels for NDF degradation in diets at tropical forage base are between 8 mg/dL. On average, the treatments showed 10.5 mg/dL, which may result in a better nitrogen supply in relation to the microbial requirements for fiber degradation [41]. Ruminal NH_3_^−^N acts as a regulator of microbial ureolytic activity, playing a fundamental role in the regulation of urea transfer to the rumen [42,43]. Although there was no difference in concentration between treatments, there was a peak of NH_3_^−^N for the PRU and U + PRU treatment in the first hours 3 h after supplementation, supported by the results found in the studies by Carvalho et al. [12] and Oliveira et al. [11]. However, our results suggest some degradation of PRU in the rumen. Poor quality forages increase NH_3_^−^N accumulation when concentrations exceed 10–12 mg/dL [5], which may be associated with energy limitations due to the low availability of forage used for absorption of microbial nitrogen [11]. The higher ammonia concentration is represented by the peaks after supplementation, according to Carvalho et al. [12]. The increase in ammonia and ruminal pH, may indicate the compromise of some processes, in which excess ammonia can be absorbed through the rumen wall; however, its return to the rumen as urea is compromised. If the concentration of urea in the blood was higher, it could be correlated with a greater return of urea to the rumen by nitrogen recycling and excretion in the urine [8], which was not observed in this study. In other words, ammonia would be returned to the ornithine cycle to resynthesis of urea, resulting in increased urea nitrogen [44].

For CONT, this peak was only observed 6 h after supplementation, a result similar to that found in the study by Oliveira et al. [11], using treatment with continuous infusion of urea in the rumen. It is expected that the concentration of NH_3_^−^N increases with the inclusion of urea, considering the high rate of degradation in the rumen, as observed in the study by Cidrini et al. [27] with supplements with low and high inclusion of urea. The peak of ammoniacal nitrogen occurred 3 h after supplementation. This later peak of conventional urea would result in an accumulation of ammonia, suggesting that microorganisms were unable to utilize N or that microbial growth was slower than nitrogen solubilization, and considering the limited energy availability of the basal diet, this would result in a breaking point [11], which may have happened with the CONT peak only 6 h after supplementation.

The increase in the concentration of NH_3_^−^N throughout the day causes a change in the microbial proportions in the rumen and, consequently, the concentration of total SCFA is modified [45]. This can be supported by the difference in total SCFA production and lower ruminal pH values throughout the day in the CONT treatment. During the dry season, due to the nutrient limiting condition of forages, there is a reduction in DM fermentation, long digest retention time, and low SCFA absorption resulting in low ADG in growing cattle. The total SCFA concentration has a strong relationship with the carbohydrate source [46], and also with the amount of DM fermented. Consequently, no change was observed in the individual molar proportion of SCFA, except for propionate.

Regarding the supplement disappearance rate, it was observed that the U + PRU treatment, due to the high-protein content coming from urea, limited the intake of animals in both experiments, and after 24 h after offering the supplement, there were leftovers in all experimental periods. The greater the inclusion of urea, the greater the time spent for the ingestion of the supplement. This intake limitation possibly generated anomalies, mainly in relation to ruminal parameters, due to collection times. The variations in ADG over the experimental periods occurred as a function of available forage and interaction with supplementation. Therefore, in the last experimental month, a great reduction in the availability and nutritional content of the forage caused a decrease in the gain of the animals in all treatments.

Diets with low availability of soluble carbohydrates, such as low-quality forage, limit the use of NPN by not providing adequate energy, resulting in slow digestion of available carbohydrates. Therefore, any positive association between the availability of non-degradable protein in the rumen and microbial use of nitrogen from recycled urea may be limited by the low availability of energy for ruminal fermentation and, consequently, for microbial assimilation of recycled nitrogen [10].

Phylum Actinobacteriota is a group of Gram-positive bacteria, and its families *Kineosporiaceae, Frankiaceae,* and *Pseudonocardiaceae* were found only in PRU and U + PRU. These groups of bacteria are capable of degrading macromolecules present in the soil, mainly proteins, starch, humus, cellulose, and lignin [47], and are involved in the organic matter cycling process [48]. Bacteria of the phylum Chloroflexi have been found in anaerobic bioreactors, soils, and aquatic environments where there is the presence of sulfur, with a function in the degradation of OM. They also act in the degradation of butyrate [49]. The diversity and distribution of these soil bacteria mainly depend on the pH, so the ability to grow in a wide range of soil pH, between 6.5–7, showed greater bacterial richness. The *Gitt-GS-136* family, present only in the PRU, shows a positive correlation with the pH value of the environment [50]. The *Ktedonobacteria* present only in U + PRU has a negative correlation with pH. This can be supported by the pH values of the two treatments, in which PRU and U + PRU show a higher pH in the first hours and PRU drops at 12 h, but U + PRU does maintain a high pH.

Meanwhile, the phylum Methylomirabilota, a poorly studied group containing the *Rokubacteriales*, was consequently found only in PRU and U + PRU, reinforcing the treatment effect. This family contains genes involved in the production and transport of lipids across the cytoplasmic membrane to the outer membrane [51]. *Rokubacteriales* encode several carbon transport proteins, including lipids, peptides, and sugars. These also contain genes involved in nitrogen respiration that can act as electron acceptors during anaerobic conditions, and nitrite oxidoreductases, which are conserved nitrification proteins [51].

Microorganisms of the genus *Methanobacterium* are hydrogenotrophic methanogenic Archaea, and play an important role in the balance in the rumen ecosystem, helping to maintain the ideal pH and using the H_2_ present in the medium, contributing to the regeneration of cofactors such as NAD+ and NADP+ [52]. There was a difference between treatments for Archaea diversity, which may indicate that this genus with greater abundance in the U + PRU is involved in the use of nitrogen by ruminants in both compartments.

The genus *Endomicrobium* was more abundant in CONT, which was also detected in soils, contaminated aquifers, termite intestines [53], and in the intestinal tract of ruminants [54]. Strains of this genus use the peptide degradation pathway, which comprises transamination reactions that form alanine, which is lost in substantial amounts [55]. These strains possibly do not participate in the intestinal digestion of plant fibers [56].

Firmicutes were the most dominant, with 46% of the total reads analyzed. The phylum included the genera Anaerovorax, Bacteroides_pectinophilus_group, Clostridia_vadinBB60_group, Uncultured, Eubacterium, Flavonifractor, Howardella, Lachnospiraceae_NK3A20_group, Lachnospiraceae_XPB1014_group, and Lysinibacillus, found in greater abundance in the PRU. In addition, Firmicutes abundance, Firmicutes: Bacteroidetes ratio, the biosynthesis of Val, Leu, and Ile, and the metabolism of linoleic acid were positively associated in the PCA mainly in the PRU group. Therefore, it is probable that these microorganisms are linked to the use of nitrogen from recycled urea in the rumen and the biosynthesis of these three branched-chain amino acids.

Some genera of this same phylum were present only in the combination of sources (U + PRU), such as *Allobaculum, Peptococcus, Staphylococcus,* and *Weissella*, which are generally found in nutrient-rich environments such as dairy products, meats, vegetables, soils, and at mammalian skin and mucosa [57]. In addition, at PCA the U + PRU group was positively correlated with the Peptostreptococcaceae family, the CP digestibility, the degradation of Lys, Val, Leu, and Ile and the metabolism of nitrogen indicating that U + PRU increases the N availability in the rumen, and consequently the ruminal microbial population and metabolic pathways keys to crude protein digestibility.

The *U29-B03* are from rumen environments, Du et al. [58], studied the effects of different dietary energy levels on the rumen bacterial population and the meat quality of the Yaks longissimus thoracic (LT) muscle. The *U29-B03* genus had a positive influence on SCFA, suggesting its participation in carbohydrate metabolism to produce SCFA, thus facilitating the deposition of intramuscular fat to promote tenderness in the LT muscle. This fact may support a higher concentration of total SCFA in the CONT in relation to the other treatments, due to the greater abundance of this genus in this treatment.

The genera of the Actinobacteria phylum play an essential role in the cycling of elements and the availability of nutrients, with the ability to decompose complex substances and the potential to be used in the degradation processes. The greatest abundance of these genera was present in PRU and U + PRU, corroborating the idea that post-ruminal urea exerted a strong influence on the dominance of these microorganisms, due to the greater availability and use of nitrogen via recycling.

Another phylum with an important role is Fibrobacterota, including *Fibrobacter*, which is one of the main cellulolytic bacteria in the rumen [59,60]. Granja-Salcedo et al. [61] observed changes in bacterial composition indicating that the succinate and propionate production pathway in the rumen was stimulated, as nitrate increased the bacteria succinate and propionate formers, such as Bacteroides and Fibrobacter. This supports the highest molar proportion of propionate in this treatment. This is in line with the negative association observed among the Fibrobacteraceae family and the Acetate: propionate ratio, observed at PCA.

The phylum Proteobacteria was influenced by the protein source. There was a greater abundance of all genera found within this phylum in the PRU and U + PRU, probably due to the higher intake and apparent digestibility of the protein. Dietary patterns characterized by a high-protein intake contribute to a higher amount of protein available, causing an increase in the concentration of protein-fermenting bacteria, especially bacterial species of the phylum Proteobacteria [62,63]. This is supported by the positive correlation observed at PCA among the Proteobacteria phylum, the crude protein digestibility, and the degradation of some amino acids such as Val, Leu, Ile, and Lys.

The ruminal abundance of the phylum Spirochaetota agrees with both reports of Wu et al. [64] and Wei et al. [65], who found a total of 1% to 1.8%, respectively, for the same phylum. *Treponema* was the only group observed at the genus level within this phylum. In the rumen, this phylum is mainly involved in the fermentation of soluble carbohydrates, using ruminal ammonia as a source of nitrogen. Higher abundances are beneficial to improve the efficiency of nitrogen conversion in the rumen [66]. CONT and PRU showed no difference in the abundance of this genus, which may be linked to the nitrogen conversion efficiency, supported by the similar ADG levels between treatments.

PRU supplementation had no effect on rumen parameters, suggesting that the direct action of abomasal supplementation seems to be based on increased availability of amino acids absorbed in the small intestine, having no direct effect on rumen fermentation [67].

From the results obtained in this study, it is observed that supplementation with conventional urea and post-ruminal urea present similar results, and different ways of improving the N status in animal metabolism depending on the nutritional characteristic of the forage and the form of absorption. However, it is known that conventional urea has its limitations. It is rapidly processed and degraded by ureolytic bacteria, increasing the concentration of NH_3_^−^N. When not used quickly, it can accumulate in the rumen environment and cause negative effects. As demonstrated in this study and the studies carried out by Carvalho et al. [12] and Oliveira et al. [11], the use of a post-ruminal delivery nitrogen source has a great potential. 

## 5. Conclusions

The use of post-ruminal urea influenced protein intake, supplement, crude protein digestibility, and the microbial population. However, it did not influence blood, rumen parameters, and animal performance. Supplementation with post-ruminal urea provided similar results to those of conventional urea. Nonetheless, the use of the combination of the two sources of urea presented a limitation in the intake of animals due to the high concentration of protein coming from urea, as well as from the high urinary excretion.

## Figures and Tables

**Figure 1 animals-13-00207-f001:**
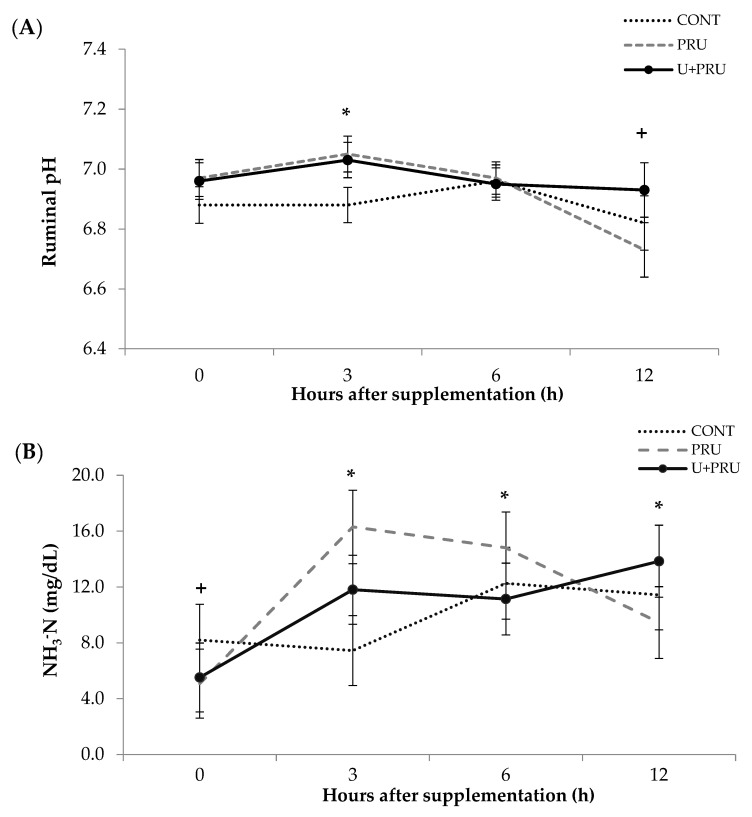
Mean and standard error means of ruminal pH (**A**) and ammonia nitrogen (NH_3_^−^N), (**B**) concentration recorded hours after supplementation, in Nellore steers supplemented with post-ruminal urea and conventional urea during the dry season. Effect of treatment and time interaction. ***** = significance (*p* ≤ 0.05), and + = tendency (*p* ≥ 0.05 and *p* ≤ 0.10). CONT: Conventional urea; PRU: Post-ruminal urea; U + PRU: Conventional urea + post-ruminal urea.

**Figure 2 animals-13-00207-f002:**
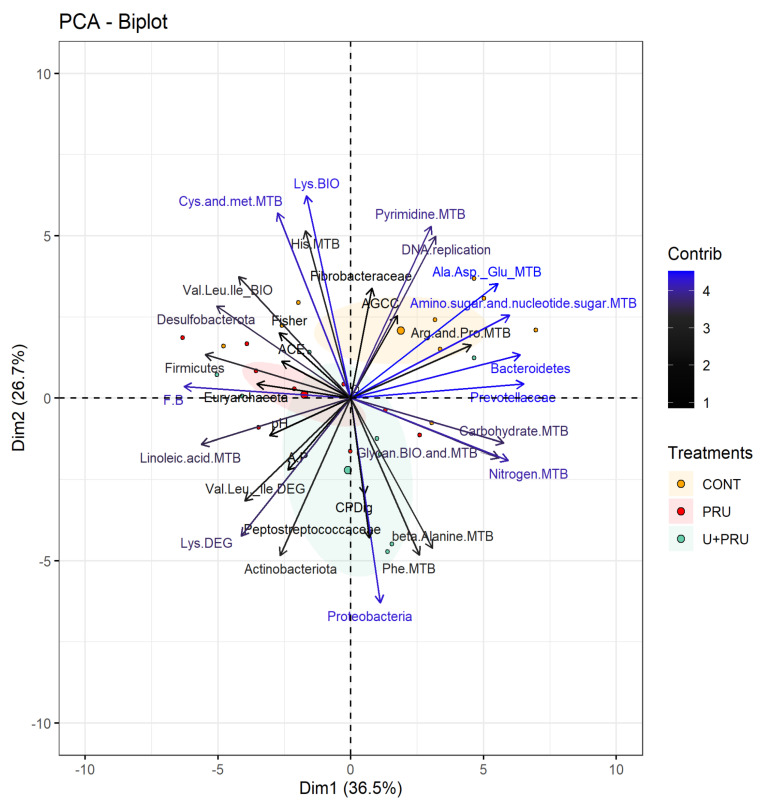
Principal component analysis (PCA) from rumen microbiota associated with KEGG pathways, fermentation parameters, and nutrient digestibility in Nellore steers as a function of the use of post-ruminal urea and conventional urea during the dry season. Ellipses represent the Euclidean distance among treatments. Blue letters represent variables with a higher contribution. MTB = metabolism, DEG = degradation, BIO = biosynthesis, Dig = Digestibility, A.P = acetate propionate ratio, F.B = Firmicutes Bacteroidetes ratio.

**Figure 3 animals-13-00207-f003:**
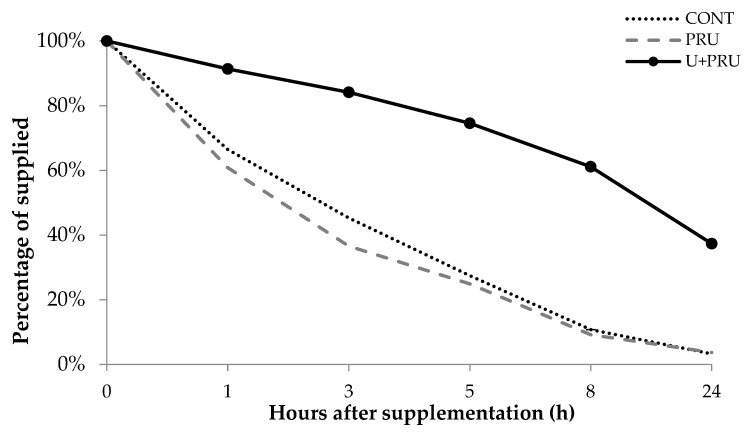
Disappearance rate of supplements as a function of the use of post-ruminal urea and conventional urea in Nellore young cattle in the dry season. (*p*-value: Treatment: <0.001; Hour: <0.001; and Treatment vs. Hour: <0.001).

**Figure 4 animals-13-00207-f004:**
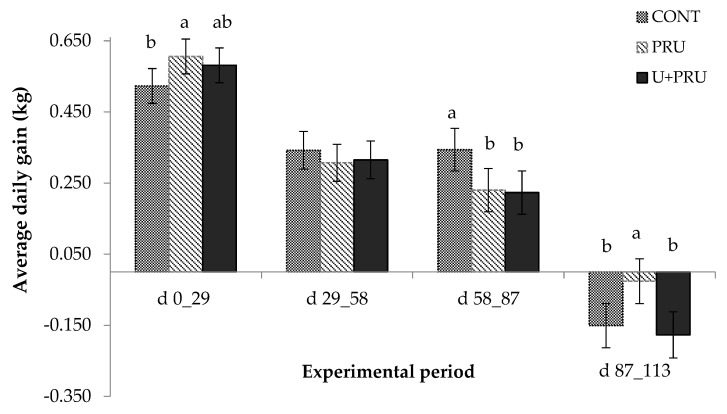
The mean and standard error means of on average daily gain in Nellore young bulls, as a function of the use of post-ruminal urea of the animals during the experimental period. Experimental days: d 0_29: First period, 29_58: Second period, 58_87: Third period, 87_113: Fourth period. Treat: Treatment; Per: Period; Treat × Per: Interaction between treatment and period. Lowercase letters on the line differ among treatments (*p* ≤ 0.05).

**Table 1 animals-13-00207-t001:** Chemical composition of supplements used.

Item	CONT	PRU	U + PRU
Ingredients (%)			
Soybean meal	36.5	36.5	36.5
Kaolin *	23	21.6	15
Ground corn	9	9	9
Urea	11.3	-	11.3
Post-ruminal urea	-	12.7	8
Minerals	20.2	20.2	20.2
Composition (% dry-matter)			
Crude Protein	50	50	70
Dry matter	88.6	88.5	86.6
Mineral matter	36.0	34.9	30.3
Ether extract	1.4	2.4	1.8
NDF	18.9	18.7	16.2
Non-protein nitrogen	32.5	32.5	58

NDF: Neutral detergent fiber. CONT: Conventional urea—Lambisk S (commercial product of Bellman-Trouw Nutrition); PRU: Post-ruminal urea; U + PRU: Combination of fonts—conventional urea and post-ruminal urea. * Formula filling.

**Table 2 animals-13-00207-t002:** Quantitative and qualitative characteristics of *Urochloa brizantha* cv. Marandu during the growing phase of Nellore steers and Nellore young bulls, supplemented with post-ruminal urea and conventional urea in the dry season.

Item	Exp. 1			SEM	Exp. 2				SEM
	CONT	PRU	U + PRU		d 0–29	d 29–58	d 58–87	d 87–113	
Quantitative Characteristics									
Height (cm)	21.0	22.0	22.1	1.940	19.7 a	21.1 b	20.1 ab	18.2 c	1.142
Forage mass (kg DM/ha)	4405	4313	4346	621.4	3762 a	3652 a	3520 a	2861 b	365.5
Density (kg DM/m^3^)	2.20	2.00	2.10	0.152	1.96 a	1.64 b	1.65 b	1.50 b	0.094
Green leaf (%)	3.20	7.27	3.85	1.856	2.78 a	2.91 a	0.0 b	0.30 b	0.844
Green stem (%)	4.40	4.30	4.10	1.759	0.52 a	0.0 a	0.0 a	3.81 b	0.993
Senescent leaf (%)	27.6	24.7	29.3	7.050	27.1 a	16.2 b	19.6 b	16.2 b	1.450
Senescent stem (%)	64.8	63.7	62.7	6.879	69.6 a	83.8 b	80.4 bc	83.5 b	1.617
Forage offer (kg DM/kg BW)	2.22	2.12	2.00	0.342	5.64 a	5.36 ab	5.02 b	4.24 c	0.509
Stocking rate (AU ha^−1^)	-	-	-	-	1.48	1.51	1.55	1.51	0.122
Qualitative Characteristics (g/kg DM)									
Dry matter	766	749	750	16.7	813 a	857 b	878 c	806 a	8.77
Crude protein	43.4	44.7	44.6	1.91	44.5 a	33.0 b	30.4 c	30.2 bc	2.59
Ethereal extract	8.72	8.57	9.69	0.87	9.30 a	5.83 b	5.78 b	4.17 c	0.45
Mineral matter	65.2	64.8	69.4	1.55	64.4 a	61.7 a	55.0 b	50.1 c	1.92
NDF	790	788	774	5.79	766 a	828 b	824 b	849 c	8.71
ADF	409	412	397	4.65	390 a	454 b	467 c	488 d	9.48
Lignin	61.1	58.7	55.1	2.17	54.6 a	60.8 a	58.1 a	66.1 b	4.47
iNDF	357	336	326	11.6	328 ab	388 b	412 b	466 c	13.6

CONT: Conventional urea (Lambisk S); PRU: Post-ruminal urea; U + PRU: Conventional + post-ruminal urea. DM: dry-matter; NDF: Neutral detergent fiber; ADF: Acid detergent fiber; iNDF: Indigestible neutral detergent fiber (iNDF). * Exp 1. (treatments); * Exp. 2 (experimental periods) 1° (First)—0 to 29; 2° (Second)—29 to 58; 3° (Third)—58 to 87; 4° (Fourth)—87 to 113. SEM: Standard error means; AU: Animal units: 450 kg body weight. No treatment effect observed (*p* > 0.05), there was a period effect in the presented variables (*p* ≤ 0.05). Lowercase letters on the line differ among treatments (*p* ≤ 0.05).

**Table 3 animals-13-00207-t003:** Intake and apparent total-tract digestibility in Nellore steers, supplemented with post-ruminal urea and conventional urea during the dry season.

Item	Treatments			SEM	*p*-Value
	CONT	PRU	U + PRU		
Intake (kg/day)					
Dry matter (DM)	9.30	10.4	9.41	0.871	0.258
Organic matter (OM)	8.52	9.43	8.55	0.808	0.296
Crude protein (CP)	0.702 b	0.874 a	0.870 a	0.060	0.005
NDF	7.13	7.73	7.04	0.670	0.392
Forage	8.62	9.37	8.66	0.821	0.454
Supplement	0.680 b	1.003 a	0.756 b	0.055	<0.001
Apparent Digestibility (g/kg)					
Dry matter	355	366	383	30.42	0.129
Organic matter	326	333	348	28.44	0.211
Crude protein	268 c	308 b	356 a	14.49	<0.001
NDF	272	273	286	23.62	0.332

SEM: Standard error means. Lowercase letters on the line differ among treatments (*p* ≤ 0.05). CONT: Conventional urea; PRU: Post-ruminal urea; U + PRU: Conventional urea + post-ruminal urea. NDF: Neutral detergent fiber.

**Table 4 animals-13-00207-t004:** Ruminal fermentation and blood parameters at different times after supplementation in Nellore steers supplemented with post-ruminal urea and conventional urea during the dry season.

Item	Treatments			SEM	*p*-Value		
	CONT	PRU	U + PRU		Treat	Hour	Treat × Hour
pH	6.88	6.93	6.97	0.054	0.232	0.006	0.039
NH_3_^−^N (mg/dL)	9.86	11.4	10.6	2.289	0.261	<0.001	0.002
SCFA (mMol)	32.6 a	27.0 b	25.8 b	2.114	0.027	0.581	0.909
SCFA (mol/100mol)							
Acetate	75.8	76.4	77.2	0.520	0.186	0.562	0.347
Propionate	14.4 a	13.9 b	13.8 b	0.268	0.049	0.107	0.426
Isobutyrate	1.06	1.24	1.03	0.509	0.904	0.245	0.313
Butyrate	4.99	4.96	4.80	0.262	0.674	0.100	0.632
Isovalerate	1.62	1.18	1.61	0.336	0.443	0.846	0.930
Valerate	1.93	2.21	1.85	0.268	0.268	0.120	0.549
A:P	5.28 b	5.50 a	5.60 a	0.114	0.015	0.098	0.657
Blood Parameters							
Uric acid (mg/dL)	1.39	1.41	1.43	0.085	0.893	<0.001	0.732
Urea (mg/dL)	45.6	49.0	50.7	5.821	0.309	0.001	0.858
Albumin (g/dL)	3.86	3.73	3.79	0.209	0.703	0.056	0.256
Creatinine (mg/dL)	2.72	2.79	2.80	0.406	0.773	0.366	0.222
Total protein (mg/dL)	14.4	14.3	14.0	0.453	0.464	0.002	0.919
AST (U/L)	71.8	64.9	73.6	5.337	0.488	0.005	0.619
GGT (U/L)	19.5	18.8	20.0	1.897	0.806	0.616	0.799

SEM: Standard error means. SCFA data were analyzed only at 0 and 12 h. A:P = acetate:propionate ratio. AST = aspartate aminotransferase; GGT = gamma glutamyl transferase. CONT: Conventional urea; PRU: Post-ruminal urea; U + PRU: Conventional urea + post-ruminal urea. Treat: Treatment, Treat × Hour: Interaction between treatment and collection hour. Lowercase letters on the line differ among treatments (*p* ≤ 0.05).

**Table 5 animals-13-00207-t005:** Characterization of nitrogen utilization in Nellore steers supplemented with post-ruminal urea and conventional urea during the dry season.

Item	Treatments			SEM	*p*-Value
	CONT	PRU	U + PRU		
N Intake, g/d					
Forage	62.1	58.9	55.5	6.138	0.555
Supplement	55.6 b	79.4 a	85.8 a	4.930	0.003
Total N intake	120.2	138.7	141.8	8.015	0.156
N excretion					
Fecal DM, kg/d	6.10	6.45	5.84	0.353	0.425
Fecal N, %	1.48	1.60	1.65	0.163	0.436
Urine, L/d	21.5 b	23.8 b	27.1 a	0.914	0.002
Urinary N, %	0.78	0.66	0.67	0.116	0.545
Fecal N, g/d	91.4	103.4	96.9	12.00	0.686
Urinary N, g/d	174.8	157.7	182.5	29.53	0.597
Total N excretion, g/d	266.0	259.6	278.1	34.58	0.802
N balance, g/d	−149.4	−119.3	−138.3	28.89	0.628
N retention g/d	−1.25	−0.80	−1.02	0.199	0.219
Microbial N, g/d	171.3	176.5	197.0	21.07	0.375

SEM: Standard error means. CONT: Conventional urea; PRU: Post-ruminal urea; U + PRU: Conventional urea + post-ruminal urea. Lowercase letters on the line differ among treatments (*p* ≤ 0.05).

**Table 6 animals-13-00207-t006:** Median and interquartile range of total operational taxonomic units (OTU), Bacteria, Archaea, richness index, diversity estimators, and phylum level in Nellore steers supplemented with post-ruminal urea and conventional urea during the dry season.

	Treatments			*p*-Value
	CONT	PRU	U + PRU	
Total OTU	91272 ± 1151	84416 ± 3705	83997 ± 9793	0.987
Bacteria	94.24 ± 2.515 ^a^	90.56 ± 4.710 ^b^	91.09 ± 2.439 ^b^	0.086
Archaea	5.751 ± 2.512 ^b^	9.433 ± 4.708 ^a^	8.899 ± 2.439 ^a^	0.086
*Richness*				
Chao 1	2659.625 ± 296	2718.667 ± 198	2661.804 ± 238	0.803
Ace	2660.349 ± 313	2713.334 ± 206	2652.033 ± 241	0.803
*Diversity*				
Fisher	500.068 ± 52.31	503.657 ± 46.29	508.211 ± 39.32	0.707
Simpson	0.996 ± 0.0029	0.996 ± 0.0014	0.996 ± 0.0003	0.782
Shannon-Wiener	9.868 ± 0.231	9.973 ± 0.214	9.938 ± 0.158	0.716
*Phylum level **				
Bacteroidota	44.60 ± 8.611 ^a^	33.45 ± 4.260 ^b^	36.30 ± 3.754 ^ab^	0.062
Fibrobacterota	0.943 ± 0.265 ^a^	0.631 ± 0.174 ^b^	0.655 ± 0.202 ^b^	0.014
Proteobacteria	0.993 ± 1.125 ^b^	2.053 ± 0.688 ^ab^	2.379 ± 3.309 ^a^	0.045
Halobacterota	1.521 ± 1.116 ^b^	2.206 ± 0.412 ^ab^	2.648 ± 0.909 ^a^	0.037
F:B	0.948 ± 0.543 ^b^	1.141 ± 0.274 ^ab^	1.197 ± 0.219 ^a^	0.099

SEM: Standard error means. CONT: Conventional urea; PRU: Post-ruminal urea; U + PRU: Conventional urea + post-ruminal urea. F:B: Firmicutes:Bacteroidetes ratio. *: Only significance or tendencies are shown. Values followed by superscript letters indicate statistical differences (*p* < 0.05) based on Kruskal–Wallis test.

**Table 7 animals-13-00207-t007:** Median and interquartile range of the relative abundance of the families in Nellore steers, influenced by post-ruminal urea in relation to conventional urea during the dry season.

Domain	Phylum	Family	Treatments			*p*-Value
			CONT	PRU	U + PRU	
Bacteria	Actinobacteriota	*Microbacteriaceae*	0.016 ± 0.042 ^b^	0.097 ± 0.048 ^a^	0.093 ± 0.314 ^a^	0.015
		*Kineosporiaceae*	NI ^b^	0.000 ± 0.045 ^ab^	0.048 ± 0.060 ^a^	0.073
		*Frankiaceae*	NI ^b^	0.000 ± 0.013 ^ab^	0.000 ± 0.015 ^a^	0.089
		*Pseudonocardiaceae*	NI ^b^	0.008 ± 0.015 ^ab^	0.014 ± 0.028 ^a^	0.063
		*Corynebacteriales*	0.014 ± 0.017 ^b^	0.039 ± 0.014 ^a^	0.020 ± 0.041 ^ab^	0.071
	Bacteroidota	*Uncultured*	0.410 ± 0.190 ^a^	0.290 ± 0.120 ^b^	0.250± 0.100 ^b^	0.013
	Chloroflexi	*Ktedonobacteraceae*	NI ^b^	NI ^b^	0.000 ± 0.011 ^a^	0.038
		*Gitt.GS.136*	NI ^b^	0.004 ± 0.014 ^a^	NI ^b^	0.058
	Desulfobacterota	*Desulfobulbaceae*	NI ^b^	0.000 ± 0.006 ^a^	NI ^b^	0.039
	Fibrobacterota	*Fibrobacteraceae*	0.943 ± 0.265 ^a^	0.631 ± 0.174 ^b^	0.655 ± 0.202 ^b^	0.014
	Firmicutes	*Clostridiaceae*	0.005 ± 0.012 ^b^	0.032 ± 0.04 ^a^	0.026 ± 0.110 ^a^	0.051
		*Planococcaceae*	NI ^b^	0.013 ± 0.022 ^a^	0.018 ± 0.015 ^a^	0.018
		*Leuconostocaceae*	NI ^b^	NI ^b^	0.004 ± 0.013 ^a^	0.026
		*Staphylococcaceae*	NI ^b^	NI ^ab^	0.004 ± 0.009 ^a^	0.021
		*Clostridia_vadinBB60_group*	0.007 ± 0.017 ^b^	0.027 ± 0.005 ^a^	0.021 ± 0.018 ^ab^	0.054
		*Eubacteriaceae*	NI ^b^	0.003 ± 0.005 ^a^	NI ^ab^	0.070
	Methylomirabilota	*Rokubacteriales_WX65*	NI ^ab^	0.000 ± 0.005 ^a^	0.000 ± 0.005 ^b^	0.096
	Planctomycetota	*Pla4_lineage*	NI ^b^	NI ^b^	0.000 ± 0.004 ^a^	0.039
		*vadinHA49*	NI ^ab^	NI ^b^	0.000 ± 0.002 ^a^	0.092
	Proteobacteria	*Rhizobiaceae*	0.029 ± 0.038 ^b^	0.060 ± 0.430 ^a^	0.100 ± 0.088 ^a^	0.003
		*Beijerinckiaceae*	0.026 ± 0.032 ^b^	0.048 ± 0.025 ^ab^	0.059 ± 0.150 ^a^	0.020
		*Comamonadaceae*	0.015 ± 0.017 ^b^	0.030 ± 0.042 ^ab^	0.049 ± 0.130 ^a^	0.013
		*Oxalobacteraceae*	0.007 ± 0.024 ^b^	0.010 ± 0.042 ^b^	0.059 ± 0.096 ^a^	0.019
		*Devosiaceae*	0.000 ± 0.002 ^b^	0.014 ± 0.043 ^ab^	0.022 ± 0.035 ^a^	0.040
		*Rhodanobacteraceae*	0.007 ± 0.010 ^ab^	0.005 ± 0.007 ^b^	0.012 ± 0.017 ^a^	0.060
		*Hyphomicrobiaceae*	NI ^b^	0.009 ± 0.004 ^a^	0.008 ± 0.017 ^a^	0.022
		*Paracaedibacteraceae*	NI ^b^	NI ^b^	0.000 ± 0.002 ^a^	0.039
		*Uncultured*	0.021 ± 0.019 ^a^	0.023 ± 0.005 ^a^	0.013 ± 0.014 ^b^	0.027
Archaea	Halobacterota	*Methanomicrobiaceae*	1.521 ± 1.116 ^b^	2.206 ± 0.412 ^ab^	2.648 ± 0.909 ^a^	0.037

NI: Not identified. SEM: Standard error means. CONT: Conventional urea; PRU: Post-ruminal urea; U + PRU: Conventional urea + post-ruminal urea. Values followed by superscript letters indicate statistical differences (*p* < 0.05) based on Kruskal–Wallis test.

**Table 8 animals-13-00207-t008:** Performance of Nellore young bulls, as a function of the use of post-ruminal urea and conventional urea during the dry season.

Item	Treatments			SEM	*p*-Value		
	CONT	PRU	U + PRU		Treat	Per	Treat × Per
Body Weight (kg)							
d 0	322	321	320	151.7	0.842	<0.001	0.668
d 29	338	338	337				
d 58	347	347	345				
d 87	357	353	351				
d 113	353	353	348				
ADG (kg/day)	0.264	0.279	0.235	0.042	0.311	<0.001	0.036

SEM: Standard error of mean; ADG: Average daily gain; CONT: Conventional urea; PRU: Post-ruminal urea; U + PRU: Conventional urea + post-ruminal urea. There was treatment and period interaction for ADG (*p* = 0.036). Treat: Treatment; Per: Period; Treat × Per: Interaction between treatment and period.

## Data Availability

Data available on request due to privacy or ethical restrictions.

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
