# Peer review of "Effect of Post-Ruminal Urea Supply on Growth Performance of Grazing Nellore Young Bulls at Dry Season"

_animals, 2023, doi:10.3390/ani13020207_

Round 1

Reviewer 1 Report (Previous Reviewer 1)

There were remarkable improvement of the re-submitted draft. I have made the decision of accept of your manuscript.

Reviewer 2 Report (Previous Reviewer 2)

I have no further comments to make. The authors have addressed my previous concerns. There are a few typographical errors in the revised text that need to be dealt with.

Reviewer 3 Report (Previous Reviewer 3)

The corrections significantly improved the manuscript. The information contained in this manuscript is relevant, as a "new" strategy for the use of non-protein N in protein supplements for cattle on tropical pasture is proposed. The experimental design was well defined. However, the definition of the third treatment is questionable, as it presents a confounding factor that corresponds to the variation of two factors at the same time, the N source and the CP content, which makes it difficult to interpret the results. But this is not a limiting factor for the acceptance of this article. In this last reading, I noticed that there are some words together throughout the text and this needs to be revised, which can be done even in the final proof of the article.

This manuscript is a resubmission of an earlier submission. The following is a list of the peer review reports and author responses from that submission.

Round 1

Reviewer 1 Report

This manuscript should be rejected. Please refer the attached file.

Author Response

Caro Revisor, 

Por favor, verifique o anexo.

Reviewer 2 Report

Please also see attached file.

Review

Animals. 1941521

General.

This paper is on an important topic and the authors have designed a series of two trials to assess the effect of post-ruminal urea supplementation on N dynamics and animal performance. The trial design is mostly appropriate and the range of measurements taken are sufficient to allow for testing of the hypothesis. One flaw in the design is the decision to provide more N in the U+PRU treatment compared to the other two treatments. No clear rationale is given for this and it confounds the results in that you are altering both the total N and the form in which that N is delivered. Some discussion about this is necessary.

In the methods section it is not always clear which of the two trials the methods refer to. I suggest either making this clear in the appropriate sections of having a separate section for each trial. The forage quality was planned to be extremely low but I was still surprised by the very low estimates of digestibility (<400 g/kg). A comment on this in the discussion would be helpful. The disappearance of supplement from the trough was markedly different among treatments, especially for the U+PRU treatment, yet no measures of disappearance were taken in Experiment 1. As rumen sampling was conducted at times relative to the provision of fresh supplement (0, 3, 6, 12 h) results would have been compromised by the amount of supplement consumed at the time of rumen sampling. This is a problem with the design that needs to be addressed in the discussion.

The standard of English is highly variable with some sections well written. However, the abstract and results sections need to be carefully rewritten with the assistance of a translation service or similar. Also, there are problems throughout that should also be attended to. I have not made specific comments to each specific issue as there are too many.

The discussion around the rumen ammonia concertation curves does not match the data shown in Figure 2. Both the Control and U+PRU appear to be increasing linearly over the 12 h of observations, while the PRU treatment peaks at 3 h then begins to decline. This is opposite to what would be expected as the whole point of the PRU was to avoid the initial peak and smooth the urea concentration in the rumen out over a longer period of time. There is no discussion of this anomaly and no discussion around the possible differences in rate of supplement intake between PRU and U+PRU seen in Expt 2 and presumably also occurring in Expt 1 (although this was not measured). Assuming the rate of disappearance from the trough in Expt 1 was similar to Expt 2, then at the 3 h sampling cattle fed U+PRU would have only eaten 15% of supplement, while PRU animals would have eaten 60%. Surely this could explain the anomalies in the rumen ammonia concentrations data (Figure 2)? Discuss.

The discussion on microbial populations in relation to treatment was exhaustive and quite out of proportion to the discussion on the remaining components of the study. As this is out of my sphere of expertise I cannot comment on the science.

Specific comments.

Please see marked copy for major issues. Moderate issues are numbered in the text and expanded on below. Minor grammatical issues were too numerous to annotate.

Please refer to numbered comments in marked copy (lack of line numbers is a deficiency that reviewers should not have to deal with!)

1.       This summary does not reflect the finding that post-ruminal urea does not deliver the expected benefits.

2.       Major issues with the English in abstract that must be addressed.

3.       I assume these were the same fecal samples as noted above with separate sub samples taken for Chromium oxide analysis?

4.       In the results you state no effect on urinary N excretion, but in the discussion you state that N loss in urine was greater for U+PRU. Please be consistent.

5.       I am not an expert in microbial characterisation so can’t comment. However this section appears to be well written in comparison with sections 3.1 to 3.4.

6.       Are you implying that the N supplementation was the reason that diets were adequate in CP?

7.       This statement is untrue as there was no significant difference between the three treatments for blood urea (P = 0.309). Please include a figure showing the change in BUN over time to support these arguments.

8.       According to Figure 2B rumen ammonia continued to increase throughout the day for CON and PRU at least to 12 h, which corresponds to the peak grazing period in a 24 h cycle. So this statement needs to be altered.

Author Response

Dear Reviewer,

Reviewer 3 Report

The objective of the study was to evaluate the effect of the use of post-ruminal urea on performance, N metabolism and the ruminal environment of Nellore cattle reared on pasture during the dry period. The subject is relevant, as it aims to provide an alternative for better protein nutrition for cattle kept on tropical pasture during the dry season. However, I believe that there were two failures in defining the experimental treatments, the first was the lack of a negative control treatment (supplement without urea), and the most serious was the definition of the third treatment (U + PRU) which proposes a strategy mixing traditional urea with post-ruminal urea. Therefore, in this last treatment there was an evident confounding, since two factors were tested at the same time, the total N content (amount of urea) and the source (two sources of urea), so how is it possible to separate the effects of these two factors? Otherwise, the work was very well structured, well conducted and the results well presented, in addition to being of great scientific contribution and potential for practical application, mainly in the comparison of the two initial treatments (normal urea x post-ruminal urea). For this reason, even knowing the limitations, I recommend publication after major review of the following issues:

1. Justify in the manuscript the scientific bases for defining the third treatment that mixed normal urea with post-ruminal urea, as well as increased the total N content.

2. Justify in the manuscript the scientific bases for not including a negative control treatment, without urea.

3. Considering the great similarity of ruminal NH3-N and blood urea results, it seems that “post-ruminal” urea degraded in the rumen in a similar way to normal urea. Why data on post-ruminal urea degradation rates were not presented. How can you ensure that the “post-ruminal” urea has not actually been degraded in the rumen? This must be presented in the manuscript.

4. I recommend careful reading of the entire manuscript to correct minor grammatical mistakes.

Specific comments:

(Note: the lines were not numbered, so in the comments that follow I will only indicate the page number).

page 1. Abstract. Replace ....”MO”... with ... “OM”

page 1. Abstract. Revise the essay to improve the connection between sentences.

page 2 Introduction. There are words together, revise.

page 2 Introduction. Consider the general comments to reinforce the introduction.

page 4. Materials and Methods. The sum of the adaptation period of 14 days plus 4 periods of 28 days does not correspond to the presented trial period of 120 days, why?

 page 4. What were the criteria for defining the number of animals in each paddock?

page 5, Table 2. For experiment 2, why were the sward characteristics not presented for each experimental treatment?

page 9. Table 3. In the results, in the text, the DMI was presented in % of BW, however the data in Table 3 is in kg/d, check....

page 10. Repeated sentence....”However, trend of treatment effect and collection time (p = 0.002), before supplementation, CONT (8.23 mg/dL) treatment tended to be greater (p = 0.082) than PRU (5.1 mg/ dL) and both similar levels of NH3-N than U+PRU. There was also treatment and hour interaction for NH3-N, (p = 0.002), before supplementation, CONT (8.23 mg/dL) treatment tended to be greater (p = 0.082) than PRU (5.1 mg/dL) and both similar levels of NH3-N than U+PRU”.

page 18. Table 8. Why were probability values ​​not presented to compare the animals' body weight at the end of the experimental periods?

Note. In the discussion, especially in the microbiology part, I recommend discussing only the most relevant points for the text to be more objective.

Author Response

Dear Reviewer,
